# Characteristics, Isolation Methods, and Biological Properties of Aucubin

**DOI:** 10.3390/molecules28104154

**Published:** 2023-05-17

**Authors:** Kartini Kartini, Michelle Abigail Irawan, Finna Setiawan, Nikmatul Ikhrom Eka Jayani

**Affiliations:** Department of Pharmaceutical Biology, Faculty of Pharmacy, University of Surabaya, Surabaya 60293, Indonesia; michelle.abigail99@gmail.com (M.A.I.); finna@staff.ubaya.ac.id (F.S.); nikmatul.ikhrom@staff.ubaya.ac.id (N.I.E.J.)

**Keywords:** aucubin, biological properties, iridoid, isolation, physicochemical properties

## Abstract

Aucubin is an iridoid glycoside widely spread in the families *Cornaceae*, *Garryaceae*, *Orobanchaceae*, *Globulariaceae*, *Eucommiaceae*, *Scrophulariaceae*, *Plantaginaceae*, and *Rubiaceae*. This review is intended to provide data on the physicochemical characteristics, isolation methods, and biological activities of aucubin and its producing plants. Aucubin is unstable and can be deglycosylated into its aglycone, aucubigenin. Various chromatographic methods (column chromatography, vacuum liquid chromatography, medium pressure liquid chromatography, and high-performance liquid chromatography) have been used together to isolate aucubin, mainly with the stationary phase C-18 and the mobile phase water–methanol solution made in gradients. In vitro and in vivo studies reveal that aucubin has a wide range of activities, including anti-inflammatory, antioxidant, anxiolytic and antidepressant, antidiabetic, antifibrotic, antimicrobial, anticancer, antihyperlipidemic, gastroprotective, cardioprotective, hepatoprotective, retinoprotective, neuroprotective, osteoprotective, and renoprotective. Even though aucubin has been extensively investigated, further research in humans is urgently needed primarily to substantiate the clinical evidence. Moreover, extensive studies on its drug delivery systems will help maximize efficacy and minimize side effects.

## 1. Introduction

For clinical and pharmaceutical practices, plants can be used in the form of crude drugs, extracts, extract fractions, and isolates. An isolate is a single compound separated from other compounds in a plant and thus has a distinguishable chemical structure. In medicine, isolates can be administered at doses accurate enough to ensure consistency in their efficacy and safety of use. In addition, isolates may act as lead compounds in developing and discovering new drugs. Active plant compounds are generally secondary metabolites, i.e., organic compounds that do not play a direct role in plant growth but as a defense against the environment, including animals and humans. Some examples of secondary metabolites are terpenoids, essential oils, alkaloids, coumarins, phenolics, and flavonoids [1]. Iridoids belong to the terpenoid group and contain a six-membered ring structure with oxygen atoms fused to a cyclopentane ring. In nature, iridoids commonly exist in the form of a glycoside. There are four major groups of iridoids: simple iridoids, iridoid glycosides, secoiridoids, and bis-iridoids [2,3].

Aucubin is a type of iridoid found in several plant families, particularly *Scrophulariaceae*, *Plantaginaceae*, and *Rubiaceae*, as their chemotaxonomic marker. Aucubin is generally present in plants’ leaves, fruits, and stems, but its presence in root wastes has also been reported [4]. Different kinds of biological activity testing, both in vitro using various cell types and in vivo using test animals in varying conditions, have been documented. These studies discovered the potential of aucubin as an anti-inflammatory, antioxidant, anxiolytic and antidepressant, antidiabetic, antifibrotic, antimicrobial, anticancer, antihyperlipidemic, gastroprotective, cardioprotective, and retinoprotective agent [5]. Moreover, numerous isolation and purification techniques have been conducted to separate aucubin from other groups of compounds and iridoid compounds. This review article provides a detailed description of the physicochemical characteristics, producing plants, isolation methods, and biological activities of aucubin. This study is a literature review that collects data and information from well-published books, the Internet, and journals. The literature search was carried out without being limited by year and it was performed using search engines on PubMed and Google Scholar as well as manual searching of other databases. The comprehensive data presented here can be used for further development of aucubin.

## 2. Physicochemical Characteristics

Aucubin (Figure 1) is an iridoid compound. Based on etymology, the term iridoid means that it is obtained from the volatile monoterpenes iridodial and iridomyrmecin, which compose the defensive secretion of Australian ants from the genus Iridomyrmex. According to its biosynthetic origin, the classical name iridoid refers to natural monoterpenoids, i.e., secondary plant metabolites characterized by a cyclopenta[c]pyranoid skeleton, also called iridane (cis-2-oxabicyclo [4.3.0]-nonane). Iridoids are frequently detected in plants as glycosides and very few as non-glycosidic compounds. Iridoids are liquid or solid compounds that are most stable at normal temperatures and have a crystalline or amorphous structure with high melting points. Most iridoids have a bitter taste. Polar iridoid glycosides are dissolve well in water and alcohols (methanol, ethanol, n-butanol). In contrast, the relative solubility of aglycones in these media is slightly poorer but increases with the number of hydroxyl groups.

Aucubin has a molecular weight of 346.33 g/mol, a melting point of 181 °C, water solubility of 3.56 × 10^−5^ mg/L at 20 °C, and a logP value of −3.49 [6,7,8]. Aucubin is a glycoside whose aglycone (i.e., aucubigenin) binds to the glucose group using an O-glycosidic bond. Phytochemical screening of aucubin in a plant can be performed with the Trim–Hill color test, and the process is as follows: First, the plant material is cut into small sizes and put into a test tube containing 5 mL of 1% HCl. After 3–6 h, 0.1 mL of the extract was collected and put into another tube containing 1 mL of Trim–Hill reagent (i.e., a mixture of 10 mL of acetic acid, 1 mL of 0.2% CuSO_4_·5H_2_O in water, and 0.5 mL of concentrated HCl). Then, the tube is heated using fire for a short time, and the solution turns blue in the presence of aucubin [9]. Ultraviolet absorption spectra can detect some properties and reactions of aucubin. Generally, aucubin neither reacts with FeCl_3_, as evidenced by the absence of color change, nor reduces Fehling’s solution. However, this substance reduces the Tollens’ AgNO_3_ reagent slowly at cool temperatures and in a few seconds when heated to 100 °C.

The UV spectrum of aucubin in MeOH has λ_max_ of 204 nm, and the IR spectrum (KBr) gives υ_max_ between 3629 and 1666 cm^−1^, indicative of O-H and C=C, respectively. Meanwhile, the LC-ESIMS spectrum of aucubin presents m/z of 368 [M + Na]^+^ calculated for C_21_H_32_O_13_. The ^1^H NMR spectrum (300 MHz, DMSO-d_6_) shows δ 4.98 (1H, d, H-1), 6.27 (1H, d, *J* = 5.8, H-3), 4.81 (1H, d, *J* = 4.3, H-4), 2.92 (1H, m, H-5), 4.70 (1H, m, *J* = 7.6, H-6), 5.62 (1H, brs, H-7), 2.92 (1H, m, H-9), 3.94 (1H, d, H-10a), 4.14 (1H, d, H-10b), 4.48 (1H, d, *J* = 7.8, H-1′), 3.12 (1H, d, H-2′), 3.41 (1H, H-3′), 2.98 (1H, dd, H-4′), 3.35 (1H, m, H-5′), 3.64 (1H, d, H-6′a), and 3.90 (1H, H-6′b), and the ^13^C NMR spectrum (75 MHz, DMSO-d_6_) shows δ 97.5 (C-1), 141 (C-3), 105.9 (C-4), 46.1 (C-5), 82.0 (C-6), 130.7 (C-7), 147.9 (C-8), 47.9 (C-9), 61.0 (C-10), 100.6 (C-1′), 75.1 (C-2′), 78.8 (C-3′), 71.6 (C-4′), 78.5 (C-5′), and 62.6 (C-6′) [10].

An X-ray diffraction (XRD) analysis of aucubin (iridoid glycoside) and its aglycone form (aucubigenin) can be used to derive information on physicochemical properties, including the orthorhombic crystal of aucubin with the space group *P*2_1_2_1_2_1_, cyclopentane and pyran ring adopting an envelope conformation, and the Glc group in the ^4^C_1_ conformation. Moreover, aucubigenin’s crystals are monoclinic, with space group *P*2_1_ and the envelope conformation of cyclopentane and pyran rings [11].

Iridoids are known to be very sensitive to treatment with acidic reagents, which hydrolyze glycosidic bonds and decompose the structure of aglycones, resulting in blue-colored products. Aucubin degradation is pH-specific and occurs in highly acidic environments. The half-life is 5.02, 5.78, and 14.84 h at a pH of 1.2, 1.6, and 2.0, respectively, but can extend to more than several days at pH levels above 3.0. Moreover, aucubin is stable in plasma for at least 6 h at 37 °C [12].

Aucubin only exhibits biological activities when the glycoside is converted into its aglycone form through deglycosylation, either in vivo or in vitro. Although the exact structure of the hydrolyzed iridoid product remains undetermined, it can be assumed that the monoterpene ring is cleaved. The glucose part of the structure needs to be hydrolyzed because aucubin is more easily transported into the cell membrane in its aglycone than in glycoside form [13].

## 3. Aucubin-Producing Plants

Aucubin is the most widespread iridoid glycoside in plants. It is commonly included in the iridoids group, which currently contains almost one thousand compounds. Bourquelot and Harissey first isolated this compound in 1905 from the leaves of *Aucuba japonica* of the family *Cornaceae*. Furthermore, aucubin has also been separated from other plant families, such as *Cornaceae*, *Garryaceae*, *Orobanchaceae*, *Globulariaceae*, *Eucommiaceae*, *Scrophulariaceae*, *Plantaginaceae*, and *Rubiaceae* [14]. Plants documented as containing aucubin are presented in Table 1. *Plantaginaceae* is particularly rich in aucubin; therefore, the compound is used as a chemotaxonomic marker for this plant family. In terms of distribution, aucubin is found in all parts of plants: flowers, seeds, fruits, leaves, stems, and roots.

## 4. Isolation of Aucubin

Currently, pure aucubin is needed for varying purposes, including standardization of raw materials and traditional medicinal products, synthesis of other iridoid glycosides [47,48], product development, and pharmacological, pharmacodynamic, and pharmacokinetic studies. Scholars have been using different separation and purification methods because aucubin is located in a different matrix of compounds in each plant or plant part. The reported yields also vary from 0.004% to 1.7%.

The isolation process generally starts with air-drying and grinding the harvested plant material into powder to increase the surface area for optimal extraction. However, some researchers chose to use fresh materials to maximize compound stability, e.g., the isolation of aucubin from *Campylanthus glaber* and *Plantago myosuros* [29,38]. This was conducted to avoid enzymatic degradation. In previous studies, it has also been proven that aucubin is easily changed due to temperature, weak base, and oxidation. Therefore, it is relatively stable if extracted and separated quickly and stored hermetically [49]. A universal solvent such as ethanol or methanol, which can dissolve various compounds with a wide range of polarities, is used in extraction from plant tissues because aucubin co-occurs with iridoids and other active compounds. For instance, aucubin was successfully isolated from the flowers of *Verbascum mucronatum* with three other iridoid glucosides (ajugol, lasianthoside I, and catalpol), two saponins (ilwensisaponin A, ilwensisaponin C), and one phenylethanoid glycoside (verbascoside) [46]. To increase yields, Yang et al. used the ionic liquid 1-Butyl-3-methylimidazolium bromide (BmimBr) to extract aucubin from *Eucommia ulmoides* [17].

After digestion at 40–45 °C, the powdered material is macerated at room temperature. Maceration is the most widely used extraction method for aucubin and may be repeated two to four times to maximize yields. Another documented method is extraction by boiling and subsequent cold maceration. Unconventional methods such as ultrasound-assisted extraction (UAE) and supercritical fluid extraction (SFE) have also been also applied to extract aucubin from *E. ulmoides* [17,50]. The ethanol- or methanol-derived extract is then concentrated or dried to produce a crude extract.

The first step for separating aucubin from other compounds is partitioning the crude extract into two immiscible liquids: ether–water, chloroform–water, dichloroethane–water, or petroleum ether–water. This is mainly to separate aucubin from chlorophyll and other lipophilic compounds. Afterward, the H_2_O phase is evaporated or lyophilized. Then the aqueous phase containing aucubin is further separated using open-column chromatography (CC), vacuum liquid chromatography (VLC), or medium-pressure liquid chromatography (MPLC). Finally, the chromatography is performed with a reverse system using the stationary phase C-18 and the mobile phase water–methanol solution made in gradients. Because aucubin is a glycoside of high polarity, it has a high affinity for the mobile than the stationary phase in the reverse system, meaning that aucubin is eluted into the mobile phase and thus separated easily. In previous research, chromatography with stationary phase polyamide and gradient mobile phase methanol–water was also used to extract aucubin from *Verbascum lasianthum*, *Veronica pectinate*, and *Plantago lagopus* [10,36,41]. In 2018, Yang et al. used adsorption by macroporous resins to isolate this compound from the fruits of *Eucommia ulmoides*. Aucubin was adsorbed onto HPD850 resins and then desorbed by eluting the column in a gradient manner using 10–80% ethanol [17]. Based on these works, the chemical structure of the isolated aucubin can be determined using UV spectrophotometry, mass spectrometry, infrared spectrometry, ^1^H NMR, ^13^C NMR, or optical rotation. Table 2 summarizes the isolation methods of aucubin from various plant species. The general steps in the isolation process of aucubin was proposed and it is presented in Figure 2.

## 5. Biological Properties

The biological activities of aucubin (Figure 3) have been extensively identified, both in vitro and in vivo. In in vivo testing, aucubin is administered via the intraperitoneal (i.p.) injection route more often than orally (p.o.). Pharmacokinetic testing proves the bioavailability of aucubin to be higher with the former (76.8%) than the latter route (19.8%). This may result from the unstable nature of aucubin in the acidic gastric juice, poor absorption onto the gastrointestinal tract due to low lipophilicity, and possible first-pass effects in the liver [12]. In in vitro assays, aucubin is tested on various animal and human cell cultures, either with or without prior induction. The biological activities of aucubin are generally dose- or concentration-dependent.

### 5.1. Anti-Inflammatory Activities

Inflammation is the body’s response to cell and tissue damage caused by various stimuli, which can be mechanical stimulants (abrasion, impact, distortion), chemical stimulants (inflammatory cytokines, chemotherapy), infections by pathogenic organisms (bacteria or viruses), etc. [52]. When tissues are exposed to and damaged by stimulants, phospholipids in the cell membrane are converted to arachidonic acid by phospholipase and then to inflammatory mediators called prostaglandins by cyclooxygenase (COX-1, COX-2) and leukotrienes by lipoxygenase (LOX). One of the signaling pathways involved in forming inflammatory cytokines (e.g., COX-2, IL-1β, IL-8, IL-10, and TNF-α) is the NF-κB pathway. Previous studies on the anti-inflammatory properties of aucubin are summarized in Table 3.

Table 3 shows that the in vitro anti-inflammatory testing of aucubin involves various types of cell lines and stimulations by different chemicals, such as LPS, IL-1β, and TNF-α. This aims to increase the release of inflammatory mediators to make the anti-inflammatory effect easily observable. In vitro testing on 3T3-L1 adipocytes with TNF-α-induced inflammation proved that aucubin could suppress the secretion of proinflammatory cytokines (MCP-1, PAI-1, and IL-6) by inhibiting the degradation of IκBα and the translocation of the p65 subunit, thus inactivating NF-κB [53]. In murine chondrocytes, aucubin inhibited the phosphorylation of IKKα/β, IκBα, and p65 (IL-1β) and p65 translocation from the cytosol to the nucleus. Both inhibitory responses prevented the expression of inflammatory mediators (MMP, iNOS, COX-2, NO, etc.) [54]. Meanwhile, in THP-1 macrophages stimulated using LPS, aucubin could impede the production of TNF-α, IL-6, IL-1β, and IFN-γ [55].

Aucubin also has the potential as an anti-inflammatory in certain pathological conditions, including diabetes, gastric mucosal lesions, and epilepsy. In vivo administration of aucubin to hyperglycemic mice decreased p-IκBα expression, accumulated p65 nuclei in the NF-κB pathway, and inhibited the expression of inflammatory cytokines (IL-1β, IL-8, IL-10, and TNF-α) [56]. In a different study, intragastrical administration of aucubin to mice with gastric mucosal lesions reduced the IL-6 and TNF-α levels in the gastric mucosa by blocking the activation of NF-κB [57]. In mice with pilocarpine-induced epilepsy, aucubin lowered proinflammatory cytokine levels (IL-1β, HMGB1, and TNF-α) [58].

Furthermore, aucubin is a promising anti-inflammatory in neurons. It inhibits the release of HMGB1 in H_2_O_2_-stimulated neurons by reducing oxidative stress, thus decreasing HMGB1-TLR4 binding, NF-κB activation and, consequently, inflammatory cytokine levels. A similar effect was also observed after the intraperitoneal administration of aucubin to male and pregnant mice with traumatic brain injury [59]. Moreover, aucubin also has anti-inflammatory potential in the eyes. In the corneal cells of humans and male rats that had their left exorbital lacrimal gland removed, aucubin inhibited the inflammatory response induced by proinflammatory cytokines by inhibiting NF-κB signals, e.g., suppressing the mRNA expression of IL-1β, IL-8, and TNF-α [60].

**Table 3 molecules-28-04154-t003:** Anti-inflammatory activities of aucubin.

No	Compound	In Vitro/In Vivo	Cell or Animal Model	Concentration/Dose	Administration Route	Ref.
1	Aucubin	In vitro	3T3-L1 adipocytes, stimulated using 10 ng/mL TNF-α	1, 3, 10, 30 µM	N/A	[53]
2	Aucubin	In vitro	Murine chondrocytes, stimulated using 10 ng/mL IL-1β	1, 10, 20, 50 µM	N/A	[54]
3	Aucubin	In vitro	THP-1 macrophages, stimulated using LPS 5 µg/mL	10, 25, 50, 100, 300 µg/mL	N/A	[55]
4	Aucubin	In vivo	Normal C57BL/6 male mice, diabetes was induced using a high-fat diet and streptozotocin	20, 40, 80 mg/kg BW	p.o.	[56]
5	Aucubin	In vivo	Male Kunming mice; gastric mucosal injury was induced using 70% ethanol	20, 40, 80 mg/kg BW	i.g.	[57]
6	Aucubin	In vivo	Mouse model of epileptic ICR with pilocarpine at 320 mg/kg BW	50, 100 mg/kg BW	i.p.	[58]
7	Aucubin	In vivo	Male BALB/c mice; induced using cisplatin	1, 5, 5 mg/kg BW	p.o. i.p.	[61]
8	Aucubin	In vitro	Neuron cells, stimulated using H_2_O_2_	50, 100, 200 µg/mL	N/A	[59]
9	Aucubin	In vivo	Male and pregnant C57BL/6 mice with traumatic brain injury	20, 40 mg/kg BW	i.p.	[59]
10	Aucubin	In vitro	3T3-L1 cells, stimulated using apoC-III	35, 70, 140, 280 µg/mL	N/A	[62]
11	Aucubin	In vivo	C57/BL6 mice, administered tyloxapol with/without aucubin using intraperitoneal injection	10, 20, 40 mg/kg BW	i.p.	[62]
12	Aucubin	In vitro	Neonatal rat cardiomyocytes, stimulated using 10 µg/mL LPS	5,15, 45 µM	N/A	[63]
13	Aucubin	In vivo	C57BL/6 mice, stimulated using LPS at 6 mg/kg BW	20, 80 mg/kg BW	Gavage	[63]
14	Aucubin	In vitro	Human corneal cells, subjected to desiccation stress	0,1, 1, 7, 15 µg/mL	N/A	[60]
15	Aucubin	In vivo	Male rats that had their left exorbital lacrimal gland removed (mouse model of dry eye disease)	75 mg/kg BW	p.o.	[60]
16	Aucubin	In vitro	Human hepatocyte HL7702 (LO2), overexpression of TLR-4	2, 4, 8, 16, 32, 64, 128, 256 µM	N/A	[64]
17	Aucubin	In vivo	Sprague–Dawley rats with IRL condition	1, 5, 10 mg/kg BW	i.p.	[64]
18	Aucubin	In vitro	RAW264.7 cells and macrophage-like THP-1 cells	50, 100 µM	N/A	[65]
19	Aucubin	In vivo	Wild-type (WT) male C57BL/6 J mice and Nrf2 knockout mice, induced using LPS	10, 20 mg/kg BW	i.p.	[65]

N/A: not applicable in the in vitro study; p.o.: per oral; i.g.: intragastric; i.p.: intraperitoneal.

Aucubin should also be considered for its anti-inflammatory effects in acute kidney failure. This compound suppressed the activation of signaling pathways involved in inflammation and apoptosis, such as NF-κB, STAT3, Akt, ERK1/2, and FOXO3a, after being orally and intraperitoneally administered to mice with cisplatin-induced kidney injury [61]. Aucubin also prevented the elevation of MMP-9, MCP-1, apoC-III, ICAM-1, VCAM-1, TNF-α, IL-1β, and IL-6 levels after stimulation by apoC-III in 3T3-L1 cells and in mice with tyloxapol-induced nonalcoholic fatty liver disease (NAFLD) [62]. In HL7702 (LO2) hepatocytes and Sprague–Dawley mice, aucubin regulated the HMGB1/TLR-4/NF-κB signaling pathway to reduce the expression of proinflammatory cytokines: TNF-α and IL-1β. Based on these results, aucubin displayed the potential as an anti-inflammatory agent in chronic liver diseases [64]. In addition to kidney failure and liver disease, this compound also shows anti-inflammatory activities in the heart. In neonatal rat cardiomyocytes (NRCMs), this compound prevented the activation of NLRP3 inflammasomes (NLRP3, ASC, pro-caspase-1) and, subsequently, reduced the expression of proinflammatory cytokines (C-caspase and IL-1β). A similar effect was observed in C57BL/6 mice administered aucubin using a gavage for 7 d [63]. The anti-inflammatory potential of aucubin in acute lung injury is supported by Qiu et al., who proved the compound’s ability to inhibit proinflammatory cytokines (IL-6, IL -18, TNF-α, IL-1β, COX2, and iNOS) and NF-κB expression in RAW264.7 and THP-1 cells, wild-type (WT) male C57BL/6 J mice, and Nrf2 knockout mice [65].

### 5.2. Antioxidant

Antioxidants are compounds that can prevent or slow down cell damage due to oxidative stress. Aucubin has demonstrated antioxidant activities in several disorders, including diabetic nephropathy, traumatic brain injury, cardiovascular disorders, liver disease, osteoarthritis, and infertility. The antioxidant activity test results of aucubin and its aglycones are summarized in Table 4.

Oxidative stress occurs when the antioxidant defense cannot balance the excess production of reactive oxygen species (ROS). This condition also refers to disruptions to the cellular redox balance. Reactive oxygen and nitrogen species originating from intracellular redox metabolism are superoxide anion radicals (O_2_^•−^), hydroxyl (OH^•^), alkoxyl and peroxyl radicals (ROO^•^), nitric oxide (NO^•^), peroxynitrite (ONOO^−^), hydrogen peroxide (H_2_O_2_), and hypochlorite (HOCl) [66]. Damage to cells, tissues, and organ systems resulting from oxidative stress is associated with a number of serious diseases, such as cancers, cataracts, neurodegenerative diseases, and even the aging process [67]. Aucubin’s potential to prevent or treat numerous diseases owing to its antioxidant properties has been confirmed using various test models.

**Table 4 molecules-28-04154-t004:** Antioxidant activities of aucubin and aucubigenin.

No	Compound	In Vitro/In Vivo	Cell or Animal Model	Concentration/Dose	Administration Route	Ref.
1	Aucubin and aucubigenin	In vitro	LX-2 cells (human hepatic stellate cell lines), induced using TGF-β1	Aucubin: 1, 10, 100, 200, 400, 800 µM Aucubigenin: 100 µM	N/A	[68]
2	Aucubin	In vitro	MC3T3-E1 (murine osteoblastic cell lines), induced using Ti particles	0.1, 1, 10 µM	N/A	[69]
3	Aucubin	In vivo	Wild-type (WT) C57BL/6 and Nrf2 knockout mice, induced using LPS	10, 20 mg/kg BW	i.p.	[65]
4	Aucubin	In vivo	C57BL/6 mice, diabetes was induced using a high-fat diet and streptozotocin	20, 40, 80 mg/kg BW	p.o.	[56]
5	Aucubin	In vivo	Kunming mice, gastric mucosal lesions were induced using 70% ethanol	20, 40, 80 mg/kg BW	i.g.	[57]
6	Aucubin	In vitro	Neuron cells, stimulated using 100 µM H_2_O_2_	50, 100, 200 µg/mL	N/A	[59]
7	Aucubin	In vivo	C57BL/6 mice, traumatic brain injury was induced using lentivirus at 4 µL	20, 40 mg/kg BW	i.p.	[59]
8	Aucubin	In vitro	3T3-L1 cells, stimulated using apolipoprotein C-III at 100 µg/mL	35, 70, 140, 280 µg/mL	N/A	[62]
9	Aucubin	In vivo	C57BL/6 mice, stimulated using tyloxapol at 300 mg/kg BW	10, 20, 40 mg/kg BW	i.p.	[62]
10	Aucubin	In vitro	Sertoli cells (primary cells and the cell line TM4), induced using 0.5 μM triptolide	5, 10, 20 µM	N/A	[70]
11	Aucubin	In vivo	Mice, induced using triptolide at 120 µg/kg BW	5, 10, 20 mg/kg BW	i.p.	[70]
12	Aucubin	In vitro	H9c2 cells, exposed to hypoxia	10, 50 µM	N/A	[71]
13	Aucubin	In vivo	C57BL/6 mice by inducing myocardial infarction	10 mg/kg BW	i.p.	[71]
14	Aucubin	In vitro	MG63 cells (human osteoblast-like cells), stimulated using dexamethasone or H_2_O_2_	1, 2.5, 5 µM	N/A	[72]
15	Aucubin	In vivo	C57BL/6 mice, stimulated using dexamethasone at 30 mg/kg BW	5, 15, 45 mg/kg BW	i.g.	[72]
16	Aucubin	In vivo	Sprague–Dawley rats with liver ischemia–reperfusion injury	1, 5, 10 mg/kg BW	i.p.	[64]
17	Aucubin	In vitro	Neonatal rat cardiomyocytes, stimulated using LPS at 10 µg/mL	5, 15, 45 µM	N/A	[63]
18	Aucubin	In vivo	C57BL/6 mice, stimulated using LPS at 6 mg/kg BW	20, 80 mg/kg BW	Gavage	[63]
19	Aucubin	In vitro	Mouse chondrocytes, induced using IL-1β	10, 20, 50 µM	N/A	[73]

N/A: not applicable in the in vitro study; p.o.: per oral; i.g.: intragastric; i.p.: intraperitoneal.

ROS, MDA, LDH, SOD, and GPx are cells’ most significant markers of oxidative stress [69]. In LX-2 cells (human hepatic stellate cell lines) stimulated using TGF-β1, aucubin and aucubigenin reduced intracellular ROS production and the mRNA expression of NOX4. Excessive ROS contributes to activating liver stellate cells (HSCs) and liver fibrosis, while NOX4 is an NADPH oxidase influencing the activation of ROS and HSCs [68]. Meanwhile, in murine osteoblastic cell cultures (MC3T3-E1) induced using Ti particles, aucubin lowered ROS, MDA, and LDH levels and increased SOD and GPx activity in cells [69]. In mouse models of diabetic nephropathy—where STZ and a high-fat diet were used to induce hyperglycemia—aucubin reduced MDA levels in kidney tissue and increased the activity of antioxidant enzymes, such as SOD, catalase, and GSH/T-GSH [56]. Lowered MDA levels and elevated SOD activities and GSH levels after the administration of aucubin were also identified in mice with induced gastric mucosal lesions [57]. In H9c2 cell cultures exposed to hypoxia and C57/BL6 mice with induced myocardial infarction, aucubin successfully reduced the NADPH oxidase subunit, P67, gp91; increased SOD and thioredoxin (Trx) levels; and lowered ROS production [71]. In rat liver homogenates with simulated IRI (liver ischemia–reperfusion injury), aucubin pretreatment significantly reduced MDA and ROS levels and elevated SOD levels, suggesting good antioxidant activity in the liver IRI model [64]. In neonatal rat cardiomyocytes (NRCMs) and C57BL/6 mice, aucubin inhibited LPS-induced oxidative stress by lowering ROS and thioredoxin interaction protein (TXNIP) levels [63]. In IL-1β-stimulated mouse chondrocytes, aucubin substantially suppressed ROS production, which is one of its mechanisms of action in treating osteoarthritis [73].

In wild-type and Nrf-2 knockout mice (i.e., model mice for studying endothelial dysfunction, oxidative stress, and microvascular attenuation), aucubin ameliorated oxidative stress by decreasing MDA and O_2_^•−^ activities and increased Nrf2-targeted signals, such as heme oxygenase-1 (HO-1) and quinone oxidoreductase-1 (NQO-1) [65]. Nrf2 is a transcription factor that can increase the transcription of numerous antioxidants and detoxification enzymes. These protective genes can quickly neutralize ROS. In neuronal cells stimulated using H_2_O_2_ on mice with traumatic brain injury, aucubin increased the translocation of cytoplasmic Nrf2 to the nucleus and the expression of antioxidant enzymes (NQO-1, HO-1, Bcl2, SOD, GSH, and GSH-Px), inhibited intercellular ROS, balanced the expression of Bcl2 and Bax, and suppressed caspase-3 activation [59]. To test the antioxidant effect on nonalcoholic fatty liver disease (NAFLD), aucubin was tested on 3T3-L1 cells stimulated using apoC-III on mice with tyloxapol-induced NAFLD. In this work, aucubin demonstrated antioxidant activities, as evident from its ability to activate Nrf2, HO-1, and PPARα and PPARγ (translocation into the nucleus), increased the phosphorylation of ACC, AMPKα, and AKT, and elevated SOD levels [62].

Oxidative stress also plays an essential part in male infertility, as it triggers apoptosis of Sertoli cells and destroys the integrity of the blood–testis barrier, thus causing testicular injury and, ultimately, spermatogenic dysfunction when exposed to pollutants or drugs. The antioxidant effect of aucubin on testicular injury was demonstrated in Sertoli cell cultures and adult male mice with triptolide-induced apoptosis. It was also confirmed that aucubin could inhibit apoptosis in the PERK/CHOP and JNK-dependent pathways by triggering Nrf2 translocation, which increased the accumulation of nucleus Nrf2 and started the expression of antioxidant enzymes in the testes and Sertoli cells [70].

Aucubin scavenges ROS directly and reduces intracellular ROS production. Aucubin also upregulates SIRT1/SIRT3, increasing FOXO3a translocation to produce antioxidant enzymes such as Mn-SOD and catalase. Generally, this effect can be detected from the low levels of malondialdehyde (MDA), which is a product of unsaturated lipid peroxidation due to ROS. In addition, aucubin can increase the translocation of Nrf2 from the cytoplasm to the nucleus. Under normal conditions, Nrf2 binds to Keap1 in the cytoplasm and triggers the translocation of the Nrf2 dimer to the nucleus, where it binds to the antioxidant response element (ARE) and induces the expression of antioxidant enzymes and protein derivatives (e.g., HO-1, NADPH, NQO-1, GPx, GST, and SOD). Aucubin regulates and balances the Bcl2-to-Bax ratio by elevating Bcl2 levels and lowering Bax levels. Bax is a proapoptotic protein, whereas Bcl2 is a protein that binds to proapoptotic proteins such as Bax.

Overall, it can be concluded that aucubin has the potential to alleviate oxidative stress in cases of liver fibrosis and nonalcoholic fatty liver disease, diabetic nephropathy, gastric mucosal lesions, myocardial infarction, liver ischemia–reperfusion injury, traumatic brain injury, osteoarthritis, and male infertility.

### 5.3. Anxiolytic and Antidepressant

Depression is a common mental health disorder, etiology and pathophysiology of which are rarely understood. There are many theories regarding the causes and mechanisms of depression, including the lack of function of the brain’s monoaminergic transmitters such as norepinephrine, 5-HT, dopamine, or their combination. Depression can be caused by decreased GABA concentrations in the cortical portions of the brain and cerebrospinal fluid [74,75,76].

Administering aucubin orally to mice at 20 and 40 mg/kg BW for 7 d proved effective in reducing anxiety. Furthermore, when administered at 10, 20, and 40 mg/kg BW, it also produced an antidepressant effect equivalent to fluoxetine [16]. The anxiolytic and antidepressant properties of aucubin are believed to result from its ability to lower glutamate levels, increase GABA levels, and inhibit monoamine oxidase A (MAO-A) and catechol-O-methyltransferase (COMT), which normally act as catalysts for the catabolism of catecholamine neurotransmitters, including dopamine, serotonin, noradrenaline, and adrenaline. However, further research is still needed to determine the real mechanism.

### 5.4. Antidiabetic

Diabetes mellitus (DM) is a chronic metabolic disease characterized by high glucose in the blood. While type 1 DM occurs due to impaired insulin synthesis and secretion (pancreatic β-cell damage), type 2 DM is caused by impaired sensitivity of the tissues (receptors) where insulin works [77]. When left unmanaged, DM can lead to various complications, such as chronic kidney disorders, retinal damage, and cardiovascular disorders.

Long-term intraperitoneal injection of aucubin can help control blood glucose levels in diabetic rats and diabetic encephalopathic rats and reduce damage to neuron cells [78]. This compound also alleviates inflammation, renal fibrosis, albuminuria, and enlargement of the glomerular extracellular matrix caused by DM by inhibiting NF-κB activation and inducing the SIRT1/SIRT3-FOXO3a signaling pathway [56]. Examining the antiglycation activity in vitro at concentrations of 0.22 mmol/L and in vivo at 10 and 25 mg/kg revealed that aucubin suppressed the formation of advanced glycation end products (AGEs). This inhibitory effect on the formation of AGEs is dose-dependent [79]. AGEs are a causative factor for, among others, chronic kidney disease and atherosclerosis. AGEs form slowly in aging, but this process is accelerated under diabetic conditions and tissue oxidation.

### 5.5. Antifibrotic

Fibrosis is a process in which fibroblasts synthesize collagen and other matrix aggregates to form scar tissue, which can disrupt organ functions because scar tissue cannot perform the role of actual parenchyma tissue [80]. Aucubin displayed a protective effect against bleomycin-induced pulmonary fibrosis in mice [81]. It reduced the mRNA expression of TGF-β1, leading to the decreased expression of TGF-β1, which regulates proliferation, extracellular matrix deposition, and the conversion of fibroblasts into myofibroblasts. In other words, aucubin inhibited the proliferation and differentiation of fibroblasts into myofibroblasts. This effect was further confirmed by the decreased levels of α-SMA, which is a key marker of myofibroblasts. In the lungs, aucubin impedes infiltration of inflammatory cells and neutrophils and reduces lactate dehydrogenase (LDH) activity. LDH is an enzyme that acts as a catalyst for converting pyruvate to lactate under anaerobic conditions [77]. The presence of lactate can activate extracellular TGF-β [82]. In addition, aucubin also reduces oxidative stress that can otherwise trigger inflammation, as characterized by low levels of MDA—i.e., a product of unsaturated lipid peroxidation by ROS [77].

### 5.6. Antifungal and Antibacterial

*Candida albicans* is a flora naturally found in the human body, especially in the mouth and teeth, throat, skin, and mucous membranes of the gastrointestinal and genitourinary tract [80,83]. *Candida* can become an opportunistic pathogen if other local normal flora and tissue health conditions that prevent the development of candidiasis decrease or an environmental imbalance occurs, such as a change in pH and nutrition. In addition, *C. albicans* can form biofilms—i.e., complex structures made up of communities of cells (e.g., hyphae, pseudohyphae, and yeast cells) attached to host tissues or surfaces, such as medical devices—as a protective mechanism for these organisms, thus complicating treatment and increasing the degree of virulence [84]. Administered at 61–244 µg/mL, aucubin exhibited an inhibitory effect on total growth, biofilm formation, metabolic activity, and cell surface hydrophobicity of *C. albicans*. Meanwhile, at 244 µg/mL, it could develop a fungicidal effect. The mechanisms involved in the antifungal activity of aucubin are still not clearly understood but are assumed to be by the inhibition of the cell surface hydrophobicity (CSH) pathway [85].

Apart from being antifungal, aucubin also shows potential as an antibacterial. This compound showed antibacterial activity on various Gram positive (*Staphylococcus epidermidis*, *S. aureus*, *Enterococcus faecalis*, and *Bacillus subtilis*) and Gram negative (*Proteus vulgaris*, *Enterobacter aerogenes*, *Klebsiella pneumoniae*, *Proteus mirabilis*, and *Citrobacter diversus*) bacteria, with MIC values from 8–128 μg/mL [86]. Moreover, aucubin extracted from *Eucommia ulmoides* leaves using the cellulase method could obviously inhibit *Escherichia coli* and *Staphylococcus aureus*, with MIC values of 9.664 and 4.832 mg/mL, respectively. However, aucubin presented weak inhibitory effect on *Streptococcus pneumonia* and *MG-hemolytic streptococcus* [87].

### 5.7. Antihyperlipidemic

Hyperlipidemia refers to a condition of elevated levels of blood lipids, such as cholesterol, triglycerides, cholesterol esters, phospholipids, and/or plasma lipoproteins, including very-low-density lipoproteins and low-density lipoproteins, and decreased levels of high-density lipoproteins. Hyperlipidemia can cause numerous medical complications, including atherosclerosis, coronary artery disease, myocardial infarction, and ischemic stroke [88]. In 3T3-L1 cells stimulated using apoC-III on C57/BL6 mice with tyloxapol-induced NAFLD, aucubin hampered the development of hyperlipidemia by influencing the regulation of ApoC-III, which controls total cholesterol, triglyceride, LDL, and VLDL levels, and by activating AMPK, Nrf2, PPARα, and PPARγ pathways, leading to the activation of fatty acid oxidation [62].

### 5.8. Anticancer

One of the targets used in cancer immunotherapy is STAT3, which is involved in cell proliferation, survival, differentiation, and angiogenesis. Under normal conditions, activation (phosphorylation) of STAT3 can release transcriptional signals from cytokine and the growth factor receptors on the plasma membrane into the nucleus [89]. However, with cancer, STAT3 becomes hyperactive and can increase cancer cells’ proliferation and survival, making it the target of cancer immunotherapy. In the cytotoxic test on human myeloid leukemia cells (KBM-5, THP-1, K562), it was found that hydrolyzed aucubin (H-aucubin) had a significant cytotoxic effect on K562 cells. In contrast, aucubin did not show any cytotoxic effects on all three cells. The cytotoxic effect of H-aucubin occurred because this compound suppressed STAT3 activation by inhibiting the protein tyrosine kinase JAK2 and c-Src, which are upstream proteins of STAT3. Aucubin’s role in suppressing STAT3 activation was also reported by Potočnjak et al. [61]. Furthermore, aucubin inhibited the phosphorylation and expression of the BCR-ABL protein in human myeloid leukemia cells, where BCR-ABL encoded tyrosine kinases—i.e., an enzyme playing an essential role in many cell functions, e.g., signaling and cell growth [90].

### 5.9. Gastroprotective

Gastric ulcers are lesions in the inner curvature of the stomach caused by *Helicobacter pylori* bacteria, use of drugs, stress, hypersecretion of HCl, etc. [80]. Administering aucubin orally at 20, 40, and 80 mg/kg BW for 3 d produced a gastroprotective effect in mice with ethanol-induced gastric mucosal lesions. The mechanism involved is the inhibition of NF-κB activation, which determines the production of proinflammatory cytokines such as TNF-α and IL-6. Aucubin increases levels of epidermal growth factor (EGF) and vascular endothelial growth factor (VEGF) to protect against ethanol-induced injuries and accelerate the healing of gastric mucosal erosion and elevates COX-1 levels, promoting prostaglandin biosynthesis that maintains gastric mucosal integrity and starting wound healing [57].

### 5.10. Hepatoprotective

The hepatoprotective activity of aucubin has been demonstrated in mice with hepatic reperfusion–ischemic injury. Administering aucubin at a dose of 5 mg/kg BW relieved liver injury through an anti-inflammatory mechanism, i.e., inhibiting the HMGB1/TLR-4/NF-κB pathway that plays a role in producing inflammatory cytokines (TNF-α, IL-1β, and HMGB1), and through an antioxidant mechanism. The latter includes elevating SOD levels, increasing mitochondrial activity, inhibiting ROS production, and reducing apoptosis through the increased expression of PGC-1α and UCP2, which can hinder ROS, and the reduced expression of caspase-3 as the cause of apoptosis [64].

Aucubin can also alleviate nonalcoholic fatty liver disease (NAFLD) caused by obesity [62] through several mechanisms: inhibiting the release of proinflammatory cytokines (TNF-α, IL-1β, IL-6) and increasing the phosphorylation of ACC, AMPKα, and AKT that stimulate more production of antioxidant enzymes. Nrf2 activation can also inhibit lipogenesis and trigger fatty acid oxidation by inhibiting the ACC1/ACC2 enzymes (catalysts for converting acetyl-CoA to malonyl-CoA), thus reducing malonyl-CoA and elevating the levels of fatty acids entering the mitochondria to be oxidized. Moreover, AMPK triggers the expression of the tricarboxylic acid cycle and increases PGC-1α expression and SOD levels.

In addition to the two conditions above, aucubin and aucubigenin also have a protective effect on hepatic fibrosis. These two compounds inhibit HSC activation by suppressing α-SMA expression, prevent over-expression of extracellular matrix (Col I, Col III) that can be deposited in interstitial cells and cause fibrosis, and decrease intracellular ROS production by reducing the mRNA expression of NOX4. NOX4 is an NADPH oxidase that determines the generation of ROS and activation of hepatic stellate cells [68].

### 5.11. Cardioprotective

In cardiac dysfunction caused by lipopolysaccharide induction, administering aucubin at 20 and 40 mg/kg BW produced a protective effect. Here, aucubin suppresses the transcription of inflammatory cytokines by inhibiting the NLRP3 inflammasomes, thereby reducing proinflammatory cytokine expression. This compound also ameliorates oxidative stress by lowering MDA, TXNIP levels, and ROS generation [63]. Furthermore, it affects the nNOS/NO pathway, alleviating oxidative stress and inhibiting ASK1/JNK signaling. With oxidative stress being reduced, the consumption of thioredoxin also decreases (Trx), causing ASK1 to form an inactive complex with Trx and prevent apoptosis in myocardial infarction [71]. In addition, aucubin also regulates Bcl2 protein expression and reduces caspase-3 activation, causing the inhibition of apoptosis. The role of aucubin in nNOS regulation was also demonstrated by other studies, where the results showed aucubin suppressing oxidative stress during cardiac remodeling and inhibiting cardiac hypertrophy due to excess pressure, hypertrophy, fibrosis, and inflammation. The cardioprotective mechanism includes the increased expression of nNOS, i.e., by regulating ion channels and thus modulating abnormal Ca^2+^ homeostasis, and mitochondrial function. nNOS expression can increase in conditions of ischemia–reperfusion injuries, infarctions, hypertrophies, and heart failures [91].

### 5.12. Neuroprotective

In research on traumatic brain injury, aucubin was reported to produce a neuroprotective effect when administered at 20 and 40 mg/kg BW as a result of its antioxidant activities (increasing Nrf2 translocation into the nucleus, activating antioxidant enzymes, suppressing ROS generation, and reducing cell apoptosis) and anti-inflammatory activities (suppressing HMGB1-mediated inflammation) [59]. In addition, aucubin at a concentration of 0.1 to 1 mM protected cells from apoptosis induced using H_2_O_2_ and facilitated neurite extension and axon regeneration in the peripheral nervous system [92]. The neuroprotective activity was also demonstrated by the ability of aucubin to prevent diabetic encephalopathy by reducing oxidative stress and inhibiting neuronal apoptosis [78].

### 5.13. Osteoprotective

Aucubin has an anti-osteoporotic effect, as it increases the expression of cytokines by differentiating osteoblasts (such as collagen I, osteocalcin, osteopontin, and osterix) in bone tissue [72]. Osteocalcin and osteopontin influence the osteoclast activity responsible for bone development and regeneration, and osterix activates osteocalcin in mature osteoblasts and regulates the final stages of bone formation. In addition, aucubin has been reported to increase the expression of antioxidative factors Nrf2 (SOD-1, HO-1, catalase), suppress ROS production, reduce Bax and cleaved caspase-3 expression, increase Bcl2 expression, and reduce the rate of apoptosis. Its osteoprotective effect is generated through anti-inflammatory and antioxidant activities. The anti-inflammatory activities include the inhibition of IKKα/β and IKBα phosphorylation and p65 subunit translocation, reducing the expression of inflammatory mediators (MMP, iNOS, COX-2, NO) and inflammatory cytokines. Meanwhile, the antioxidant activities are the inhibition of IL-1β-induced apoptosis of chondrocytes by increasing Bax expression, reducing Bcl2 expression, and suppressing ROS generation [54,73].

### 5.14. Renoprotective

Cisplatin is an anticancer compound with various side effects, including nephrotoxicity. The activity of aucubin against cisplatin-induced acute kidney injuries has been tested; the results showed that administering aucubin orally and intraperitoneally at 1.5 and 5 mg/kg BW could reduce tissue changes in the kidney. The renoprotective mechanisms involve suppressing the expression of p65 NF-κB in proximal tubular cells and TNF-α expression in the kidney, suppressing STAT3 activation that can lead to dose-dependent apoptosis, increasing the expression of antioxidant genes such as HO-1, and reducing the lipid peroxidation product 4-HNE based on the dose administered. Aucubin also suppresses the increased expression of FOXO3a, thus preventing cisplatin-induced renal cell apoptosis [61].

The renoprotective effect of aucubin can also be seen from its ability to relieve albuminuria and contain the expansion of the glomerular extracellular matrix, renal fibrosis, and inflammation caused by diabetes [56]. Aucubin inhibits the expression of p-IκBα and inflammatory cytokines (IL-1β, IL-8, IL-10, and TNF-α) and p65 nucleus accumulation. Furthermore, it upregulates SIRT1/SIRT3, causes deacetylation, increases FOXO3a translocation, and intensifies the activity of the Nrf2 signaling pathway. FOXO3a transcription produces Mn-SOD and CAT, and Nrf2 transcription generates NQO1 and HO-1; all four compounds can inhibit ROS and provide antioxidant effects.

### 5.15. Retinoprotective

Because of its anti-inflammatory and antioxidant activities, the retinoprotective effect of aucubin is expected. The mechanism involved is the inhibition of the NF-κB signal, which can suppress the mRNA expression of IL-1β, IL-8, and TNF-α and regulate the expression of the Bcl2 protein group (as a regulator of apoptosis), thus inhibiting apoptosis [60]. In another work, administering oral aucubin at a dose of 15 mg/kg BW prevented retinal degeneration induced using N-methyl-N-nitrosourea in mice. This effect is believed to result from the inhibition of oxidative damage and apoptosis of DNA, as shown by the decrease in 8-OHdG generally formed due to the generation of excess ROS [79].

## 6. Safety and Toxicity

Various studies have successfully demonstrated the biological activity of aucubin. Unfortunately, toxicity tests on aucubin have not been widely carried out. On the other hand, safety is one of the mandatory requirements for a compound to be used as a medicine. However, Chang et al. reported the results of the toxicity test on mice [93]. A series of aucubin doses (100, 300, 600, and 900 mg/kg BW) were administered i.p. in mice to measure its lethal dose. None of the test animals died after 24 h, but the activity of serum GOT and alkaline phosphatase decreased slightly at doses of 300 to 900 mg/kg BW, and triglyceride levels increased slightly. To determine the acute toxicity, three dose levels of aucubin (20, 40, and 80 mg/kg BW on mice) were administered intraperitoneally. There was no change in the enzyme activity (alkaline phosphatase, GOT, GPT) and the levels of triglycerides, glucose, urea nitrogen, and total protein in the test groups compared to the control group. Therefore, it can be concluded that aucubin is a compound with low toxicity and its lethal dose is estimated to be above 0.9 g [93,94]. Further toxicity tests, especially sub-chronic and chronic toxicity tests, are needed to determine the safety of aucubin in long-term use.

## 7. Conclusions

Aucubin is an iridoid glycoside widely distributed in the families *Cornaceae*, *Garryaceae*, *Orobanchaceae*, *Globulariaceae*, *Eucommiaceae*, *Scrophulariaceae*, *Plantaginaceae*, and *Rubiaceae*. Various isolation methods have been tested and developed, considering that pure aucubin plays an essential role in standardizing raw materials and traditional medicinal products, synthesizing other iridoid glycosides, product development, and pharmacological, pharmacodynamic, and pharmacokinetic studies. Aucubin shows promise for a variety of therapeutic and biomedical applications because of its biological activities as an anti-inflammatory, antioxidant, anxiolytic and antidepressant, antidiabetic, antifibrotic, antifungal and antimicrobial, anticancer, antihyperlipidemic, gastroprotective, cardioprotective, hepatoprotective, retinoprotective, neuroprotective, osteoprotective, and renoprotective agent. Further research in human application is urgently needed to substantiate the clinical evidence of aucubin. Sub-chronic and chronic toxicity tests are indispensable in determining the safety of aucubin in long-term use. In addition, research related to drug delivery systems must be conducted considering the unstable nature of aucubin.

## Figures and Tables

**Figure 1 molecules-28-04154-f001:**
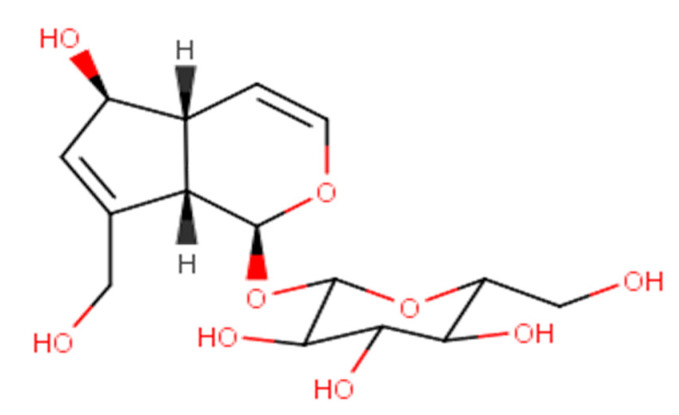
Chemical structure of aucubin.

**Figure 2 molecules-28-04154-f002:**
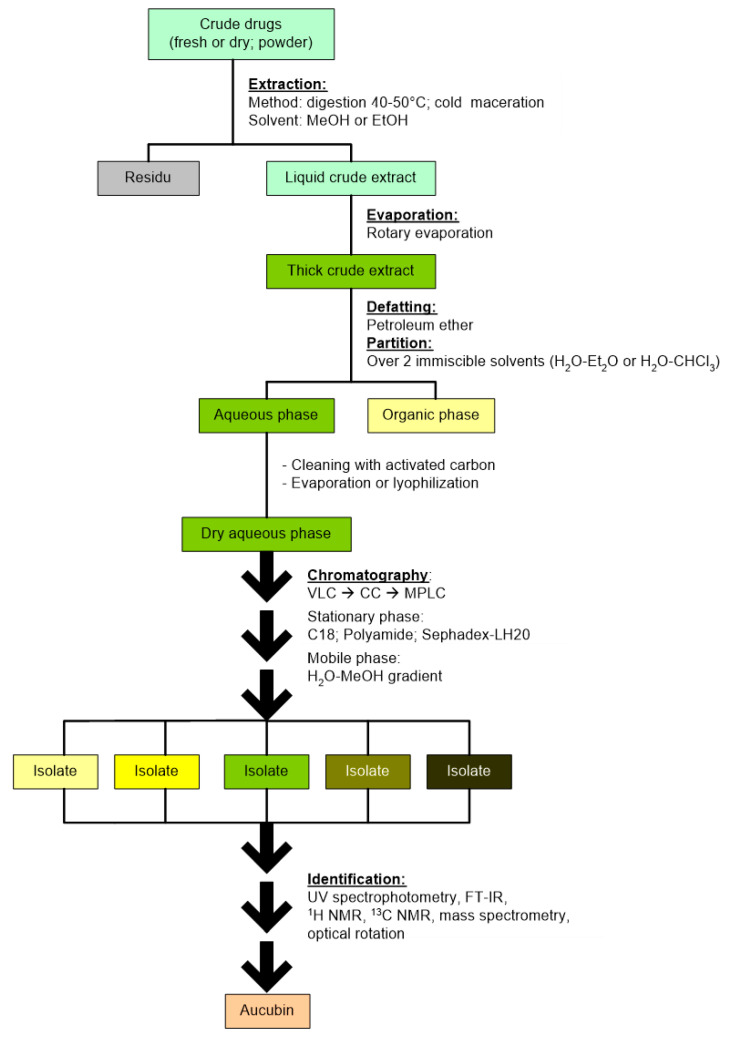
The general steps in the isolation process of aucubin.

**Figure 3 molecules-28-04154-f003:**
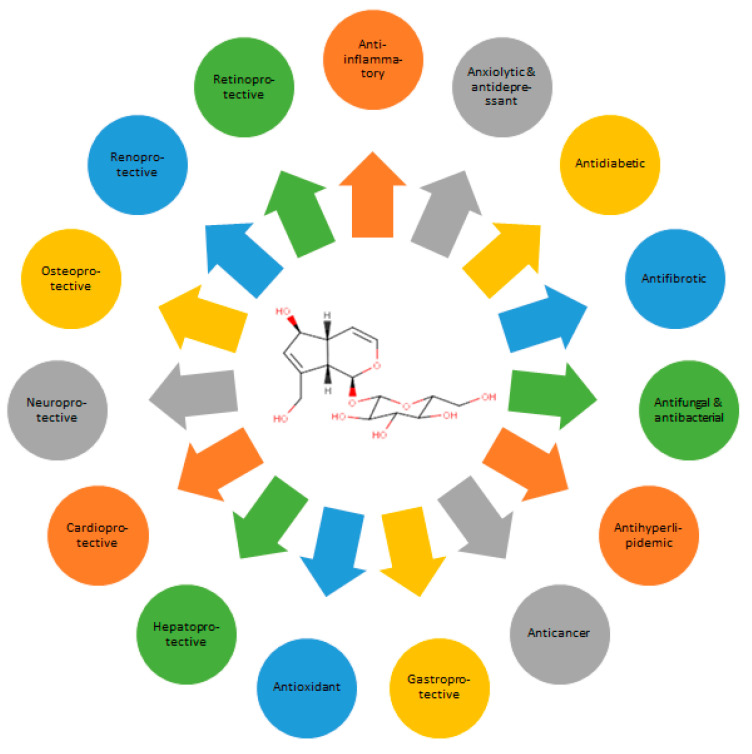
Spectra of the biological activities of aucubin.

**Table 1 molecules-28-04154-t001:** Aucubin-producing plant species.

Family	Species	Part of Plant	Reference
*Cornaceae*	*Aucuba japonica*	Leaves	[14]
*Eucommiaceae*	*Eucommia ulmoides*	Seeds, fruits	[15,16,17]
*Buddlejaceae*	*Buddleja globosa*	Leaves	[18]
	*Buddleja asiatica*	Aerial flowering parts	[19]
*Lamiaceae*	*Vitex agnus-castus*	Flowers, leaves, twigs	[20]
*Lentibulariaceae*	*Utricularia australis*	N/A	[21]
*Orobanchaceae*	*Bellardia trixago*	Aerial parts	[22]
	*Castilleja tenuiflora*	Aerial parts	[23]
	*Centranthera grandiflora*	Roots, stems, leaves, flowers	[24]
	*Melampyrum arvense*	Aerial parts	[25]
	*Parentucellia viscosa*	Whole plants	[26]
	*Rehmannia glutinosa*	Roots, leaves	[27]
*Plantaginaceae*	*Aragoa cundinamarcensis*	Aerial parts	[28]
	*Campylanthus salsaloides*	Aerial parts	[29]
	*Campylanthus glaber*	Aerial parts	[29]
	*Globularia alypum*	Leaves, flowers, woody stems, underground parts	[30]
	*Globularia dumulosa*	Aerial parts	[31]
	*Globularia cordifolia*	Roots, rhizomes	[30,32]
	*Globularia meridionalis*	Leaves, flowers, woody stems, underground parts	[30]
	*Globularia punctata*	Leaves, flowers, woody stems, underground parts	[30]
	*Linaria alpina*	Aerial parts	[33]
	*Paederota lutea*	Whole plants	[34]
	*Plantago lanceolata*	Aerial parts	[35]
	*Plantago lagopus*	Aerial parts	[36]
	*Plantago major*	Aerial parts	[37]
	*Plantago myosuros*	Whole plants	[38]
	*Veronica beccabunga*	Leaves	[39]
	*Veronica hookeri*	N/A	[40]
	*Veronica pectinata*	Aerial parts	[41]
	*Veronica pinguifolia*	N/A	[40]
*Scrophulariaceae*	*Scrophularia nodosa*	Leaves, flowers, stems, roots	[42]
	*Sutera dissecta*	Aerial parts	[43]
	*Verbascum lasianthum*	Flowers, roots	[10,44]
	*Verbascum macrurum*	Aerial parts	[45]
	*Verbascum mucronatum*	Flowers	[46]

N/A: not mentioned in the source document.

**Table 2 molecules-28-04154-t002:** Aucubin isolation methods from different plant species.

No	Plant and Plant Part	Extraction Method and Solvent	Isolation Method	Yield (%)	Ref.
1	*Eucommia ulmoides*; seeds	Smashing tissue extraction using methanol	The crude extract was defatted using petroleum ether; then, column chromatography of the residue was conducted using Si gel as stationary phase and petroleum ether-EtOAc (50:1 to 1:10) as the mobile phase, followed by another column chromatography using Sephadex LH-20 as the stationary phase and petroleum ether-EtOAc (1:8) as the mobile phase.	0.28	[16]
2	*Plantago major;* aerial parts	Maceration using methanol	The crude extract was partitioned using dichloroethane-H_2_O; the water-soluble part was then cleaned using charcoal, followed by CC using stationary phase C-18 and different mobile phases: H_2_O, H_2_O-MeOH (95:5, 70:30, 50:50), MeOH, MeOH-Me_2_CO (1:1), and MeOH-Cl(CH_2_)_2_Cl (1:1). Then, the MeOH-Cl(CH_2_)_2_Cl (1:1) fraction was purified with Si gel.	0.055	[37]
3	*Campylanthus salsaloides*; dried and fresh aerial parts	Boiling in ethanol for 5 min, followed by 6 d of maceration	The crude extract was partitioned in Et_2_O-H_2_O; the aqueous phase was then evaporated and treated with charcoal, followed by reversed-phase CC (C-size Lobar^®^) using mobile phase H_2_O-MeOH (1:0 to 2:1).	0.15 (dried aerial parts), 0.32 (fresh aerial parts)	[29]
4	*Globularia dumulosa*; aerial parts	Digestion using methanol at 45 °C	The crude extract was partitioned in H_2_O-CHCl_3_; then, the water fraction was lyophilized, followed by VLC with stationary phase C-18 and different mobile phases: H_2_O, H_2_O-MeOH (5–80% MeOH in H_2_O), and MeOH. The subsequent VLC used Si gel as the stationary phase and CHCl_3_-MeOH-H_2_O (90:10:1 to 50:50:5) as the mobile phase, followed by MPLC using stationary phase C-18 and mobile phase MeOH in water (0–40%).	0.079	[31]
5	*Aragoa cundinamarcensis*; aerial parts	Boiling in EtOH, followed by maceration for 3 d	The crude extract was partitioned in Et_2_O-H_2_O; the aqueous phase was then cleaned using activated carbon in MeOH, followed by CC using stationary phase C-18 and mobile phase H_2_O-MeOH (1:0 to 2:1).	1.7	[28]
6	*Verbascum lasianthum*; roots	Digestion using methanol at 40 °C	The crude extract was partitioned in CHCl_3_-H_2_O; the aqueous phase was then lyophilized, followed by CC using polyamide as the stationary phase and H_2_O and an H_2_O-MeOH mixture as the mobile phase. Then, VLC was conducted using C-18 as the stationary phase and H_2_O-MeOH as the mobile phase (0–100%, gradient).	0.06	[10]
7	*Verbascum mucronatum*; flowers	Digestion using methanol at 40 °C	The crude extract was partitioned in CHCl_3_-H_2_O, followed by CC using polyamide as the stationary phase and H_2_O and an H_2_O-MeOH mixture as the mobile phase. Then, VLC using stationary phase C-18 and gradient mobile phase H_2_O-MeOH (0–100%) was conducted.	0.02	[46]
8	*Plantago myosuros*; whole plants, frozen	Maceration using ethanol	The crude extract was partitioned using Et_2_O-H_2_O; the aqueous phase was then cleaned using activated carbon in MeOH, followed by CC using stationary phase Lobar RP_18_ and mobile phase H_2_O-MeOH (25:1 to 1:1).	0.04	[38]
9	*Eucommia ulmoides*; fruits	UAE using 0.5 mol/L ([Bmim]Br) ionic liquid	The ionic liquid extract was placed onto a glass column containing HPD850 resins; then, the column was washed using deionized water and eluted (desorption) using 10–80% EtOH. This process ended with the vacuum distillation of the eluent, 40–80% ethanol.	N/A	[17]
10	*Globularia cordifolia*; roots and rhizomes	Digestion using methanol at 45 °C	The crude extract was partitioned using H_2_O-CHCl_3_; then, the aqueous phase was lyophilized, followed by VLC using LiChroprep C-18 as the stationary phase and H_2_O and a mixture of H_2_O-MeOH (10–90% MeOH) as the mobile phase. The subsequent MPLC was performed using C-18 as the stationary phase and MeOH in H_2_O (0–50%, MeOH) as the mobile phase, followed by CC using stationary phase Si gel and mobile phase CH_2_Cl_2_-MeOH-H_2_O (70:30:3).	0.004	[32]
11	*Bellardia trixago*; aerial parts	Remaceration using methanol	The crude extract was partitioned using water–petroleum ether, followed by chloroform and n-butanol. The butanol fraction was then column-chromatographed using stationary phase Si gel and mobile phase CHCl_3_-MeOH-H_2_O (80:20:1, 80:20:2, to 50:50:5). Fraction D underwent MPLC using 15–25% MeOH as the mobile phase.	0.06	[51]
12	*Veronica pectinata* L. Var. *glandulosa;* aerial parts	Digestion using MeOH at 40 °C	The crude extract was partitioned using water-CHCl_3_. The water fraction was then lyophilized, followed by CC using stationary phase polyamide and mobile phase H_2_O-MeOH, made in gradients by increasing the MeOH concentration to produce five fractions. Fraction A was then chromatographed using stationary phase Si gel and mobile phase CHCl_3_:MeOH:H_2_O (90:10:1 to 60:40:4), followed by MPLC using stationary phase RP-18 and gradient mobile phase MeOH (20–50%).	0.027	[41]
13	*Paederota lutea*; whole plants	Brought to a boil using EtOH, followed by 7 d of maceration	The crude extract was partitioned using H_2_O-Et_2_O. The water fraction was then chromatographed using stationary phase RP-18 and mobile phase H_2_O-MeOH (25:1 to 1:1).	0.349	[34]
14	*Vitex agnus-castus*; flowers, leaves, and twigs	Digestion using MeOH at 45 °C	The crude extract was partitioned using H_2_O-CHCl_3_, followed by n-BuOH. The n-BuOH fraction was then column-chromatographed using stationary phase Si gel and mobile phase CHCl_3_ (by increasing MeOH gradually). Further separation and purification were conducted by CC using Si gel as the stationary phase and EtOAc:MeOH:H_2_O (100:5:2 to 100:17:13) and CHCl_3_:MeOH:H_2_O (90:10:1 to 60:40:4) as mobile phases, CC using stationary phase Sephadex LH-20 and mobile phase MeOH, and CC using stationary phase RP-18 and mobile phase MeOH in H_2_O (made in gradients).	0.006	[20]
15	*Verbascum lasianthum*; flowers	Digestion using MeOH at 40 °C	The crude extract was partitioned using H_2_O-CHCl_3_. The water phase was lyophilized and then processed with VLC using polyamide as the stationary phase and H_2_O as the mobile phase (with increasing MeOH concentrations), VLC using stationary phase C-18 and mobile phase H_2_O-MeOH (0–100% MeOH). Further separation was conducted by CC using Si gel as the stationary phase and CHCl_3_, CHCl_3_:MeOH (95:5), and CHCl_3_:MeOH:H_2_O (70:30:3) as mobile phases, and VLC with C-18 as the stationary phase and H_2_O and gradient MeOH-H_2_O (0–20% MeOH) as mobile phases.	0.028	[44]
16	*Castilleja tenuiflora*; aerial parts	Maceration using ethanol	The crude extract was partitioned using H_2_O-Et_2_O. The H_2_O phase was then concentrated, dissolved in MeOH, and cleaned using activated carbon. Further separation was carried out using CC (stationary phase Si gel and mobile phase hexane-CH_2_Cl_2_-AcOEt-MeOH with increasing polarity) and MPLC (C-18 as the stationary phase, H_2_O:MeOH (10:0 to 1:1) as the mobile phase).	0.46	[23]
17	*Plantago lagopus*; aerial parts	Digestion using MeOH at 40 °C	The crude extract was partitioned using water–petroleum ether. The H_2_O phase was column-chromatographed (polyamide as the stationary phase, 0–100% MeOH as the mobile phase). The water fraction was then extracted using n-butanol, followed by MPLC and CC (Si gel as the stationary phase; CHCL_3_:MeOH at 100:0, 95:5, 90:10, 85:15, 80:20; and 75:25 as the mobile phase).	N/A	[36]
18	*Parentucellia viscosa*; whole plants	Remaceration using 96% EtOH	The crude extract was column-chromatographed using Si gel as the stationary phase and n-butanol saturated with water and CHCl_3_/MeOH at various ratios as the mobile phase.	0.16	[26]
19	*Veronica beccabunga*; leaves	Brought to boil using EtOH	The crude extract was partitioned using Et_2_O-H_2_O; then, the H_2_O phase was dried and dissolved in 10% acetic acid. The aliquots were then column-chromatographed with stationary phase RP-18 and mobile phase H_2_O-MeOH (1:0 to 1:1).	0.25	[39]
20	*Veronica hookeri* and *Veronica pinguifolia*; N/A	Maceration using MeOH	The crude extract was partitioned using Et_2_O-H_2_O; then, the H_2_O phase was dried and column-chromatographed with stationary phase RP-18 and mobile phase H_2_O-MeOH (25:1 to 1:1).	0.18 (*V. hookeri*), 0.08 (*V. pinguifolia*)	[40]

N/A: not mentioned in the source document.

## Data Availability

Not applicable.

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
