# Peer review of "Characteristics, Isolation Methods, and Biological Properties of Aucubin"

_molecules, 2023, doi:10.3390/molecules28104154_

Round 1
Reviewer 1 Report
In this review, the introduction is comprehensive, the Physicochemical characteristics is clear, and the Biological properties of aucubin also very thorough, For all these reasons, I can recommend the acception of the manuscript after minor revision:
1. I think that information about Aucubin-producing plants could be extended, more examples should be added. This would be valuable for later publication citation.
2. The superiority of use fresh materials to maximize compound stability in the isolation of aucubin should be more emphasized.
3. The manuscript should be extended in scientific discussion. The authors presented their results and compared to some works, but did not present explanations for the reasons to reach these results.
Reviewer 2 Report
The present review manuscript entitled “Characteristics, Isolation Methods, and Biological Properties of Aucubin” authored by Kartini et al. describes the physicochemical characteristics, isolation methods, biological activities of aucubin, and its producing plants. Aucubin is unstable and can be deglycosylated into its aglycone, aucubigenin. The future prospective of the study is to carry out in humans is urgently needed primarily to substantiate the clinical evidence. Also, extensive studies on its drug delivery systems will help maximize efficacy and minimize side effects. It is a well-organized review article and lacks major errors. I would like to point out that I am impressed by the work done and the quality of the manuscript. The review was presented in a very good and accurate way. The methodology section is at an appropriate scientific level. In addition, the work is written very carefully, and the English language is good. Therefore, I recommend it for publication. However, certain Minor issues are detailed below which need to be addressed before its final acceptance in Molecules.
I advise the authors to take the following points into account while revising their manuscript.
Comment 1: Firstly, I would like to draw the attention of the authors that I found there are some typographical errors in the manuscript, so authors need to correct them in the revised manuscript.
Comment 2: Let the author focus main points and explain the research question clearly. In the abstract authors mentioned that the chromatography method is used to isolate Aucubin but practically this is not possible only with the HPLC method. Authors, please explain. So the abstract section should be revised.
Comment 3: Family species names should be in italics check though out the manuscript
Comment 4: Authors did not mention what kind of review this study falls? Literature review or Systamic review?
Comment 5: Authors mentioned only anti-fungal activity, and did not mention any anti-bacterial activity where there are many articles published on antibacterial activity
Comment 6: In Table 2 too much information is added when already given reference you can minimize the information and make a better-structured table.
Comment 7: Authors did not include any cytotoxic studies or any information as in the whole review authors targetting aucubin use as a drug.
Comment 8: Abbreviations missing in the manuscript.
Comment 9: Include the Graphical Abstract in the revised manuscript to attain a broad readership.
Comment 10: Revise the conclusion section with future prospectives and also conclusions section should be elaborated.
Comment 11: The homogeneity of the reference section needs to be maintained. In some references journal names are written in full form, and in some references journal names are in abbreviation form. So please check and revise accordingly to the journal's instructions.
